# Ultrasensitive antibody-aptamer plasmonic biosensor for malaria biomarker detection in whole blood

Antonio Minopoli [1,2], Bartolomeo Della Ventura[2], Bohdan Lenyk [1,3], Francesco Gentile[4], Julian A. Tanner [5], Andreas Offenhäusser[1], Dirk Mayer [1✉] & Raffaele Velotta [2✉]

Development of plasmonic biosensors combining reliability and ease of use is still a challenge. Gold nanoparticle arrays made by block copolymer micelle nanolithography (BCMN) stand out for their scalability, cost-effectiveness and tunable plasmonic properties, making them ideal substrates for fluorescence enhancement. Here, we describe a plasmon-enhanced fluorescence immunosensor for the specific and ultrasensitive detection of *Plasmodium falciparum* lactate dehydrogenase (*Pf*LDH)—a malaria marker—in whole blood. Analyte recognition is realized by oriented antibodies immobilized in a close-packed configuration via the photochemical immobilization technique (PIT), with a top bioreceptor of nucleic acid aptamers recognizing a different surface of *Pf*LDH in a sandwich conformation. The combination of BCMN and PIT enabled maximum control over the nanoparticle size and lattice constant as well as the distance of the fluorophore from the sensing surface. The device achieved a limit of detection smaller than 1 pg/mL (<30 fM) with very high specificity without any sample pretreatment. This limit of detection is several orders of magnitude lower than that found in malaria rapid diagnostic tests or even commercial ELISA kits. Thanks to its overall dimensions, ease of use and high-throughput analysis, the device can be used as a substrate in automated multi-well plate readers and improve the efficiency of conventional fluorescence immunoassays.

[1] Institute of Biological Information Processing (IBI-3), Bioelectronics, Forschungszentrum Jülich, 52425 Jülich, Germany. [2] Department of Physics "E. Pancini", University of Naples "Federico II", Via Cintia 26, 80126 Naples, Italy. [3] Department of Physics, University of Konstanz, 78457 Konstanz, Germany. [4] Department of Experimental and Clinical Medicine, University Magna Graecia, 88100 Catanzaro, Italy. [5] School of Biomedical Sciences, University of Hong Kong, Hong Kong SAR, China. ✉email: dirk.mayer@fz-juelich.de; rvelotta@unina.it

Fluorescence-based techniques are among the most widespread in the fields of biotechnology, life sciences, and biomedical research[1–3]. Fluorescent probes and fluorescent-labeled bioreceptors are extensively employed in optogenetic studies[4], cytogenetic assays, in fluorescent in situ hybridization[5], bioimaging analysis, for the observation of both the location and motion of cells or subcellular elements[6], molecular dynamic investigation[7], as well as in fluoroimmunoassays for the detection of molecular biomarkers and in enzyme-linked immunosorbent assay (ELISA)[8]. However, the poor yield of many fluorescent dyes and signal interference can hamper the application of fluorescence to the field of sensing. The detection of analytes in very low abundance ranges is a long-standing goal of fluorescence. Many efforts have been carried out to enhance the fluorescent signal in order to extend their application to ultra-sensitive fluoroimmunoassays[9,10].

Plasmonic nanostructures are realistic candidates to extend the limit of fluorescence detection to the femtomolar level and beyond[11,12]. These devices modify the spectral properties of the nearby fluorescent dyes and do not necessitate expensive equipment, specific or toxic reagents, or significant modifications to well-established fluorescence-based assays. Such a modification depends strongly both on the spectral overlap between the fluorescent dye and the plasmon absorbance[13,14] and on the fluorophore-nanostructure distance $z$[15–17]. In particular, if the plasmon absorbance overlaps the fluorophore excitation peak, a strong fluorescence enhancement (FE) is achieved for small values of $z$ (<10 nm) (up to 100-fold)[17,18], thanks to the Förster resonance energy transfer (FRET) mechanism, whereas a weak coupling occurring at higher values of $z$ leads to a vanishingly small FE[14,16]. In contrast, the overlap with the fluorophore emission peak provides a large FE for high $z$ (>10 nm), thanks to the enhancement of the fluorophore radiative rate through the Purcell effect, while a progressive decrease of the FE occurs at smaller $z$'s, due to the increasing fluorescence quenching via non-radiative losses in the metal nanostructure[14,16]. In the case that the plasmon absorbance superimposes over both the excitation and emission peak of the fluorophore, i.e., dual-mechanism enhancement[19], a huge FE arises at an optimal distance of ~10–15 nm, a strong quenching being present at shorter $z$'s, whereas a return to the no enhanced fluorescence conditions occurs at longer distance due to the weaker plasmon-fluorophore coupling[14,16–18].

Two-dimensional (2D) arrays of metal nanostructures are particularly suitable as plasmon-enhanced fluorescence (PEF)-based biosensing platforms[12]. Until now, a variety of fluorescence enhancers, such as gold micro-islands[20], arrays of metal nano-objects (e.g., nanoparticles[21], nanorods[22], nanotriangles[23], nanocrystals[24]), bow-tie nanoantennas[25] and resonant nanocavities[26,27], have been exploited to push down the detection limit in fluorescence-based assays. Despite significant milestones in terms of limit of detection (LOD) (down to fM level[28,29]), FE factor (up to $10^5$-fold[27,30]) and adaptability to the conventional and well-established fluoroassays[31,32], still a number of factors limit the use of those solutions as devices for routine testing, point-of-care analysis and large-scale use of the PEF-based fluoroassays, including, to cite a few, the need for skilled personnel, expensive technologies, time-consuming procedures, poor versatility in tuning of plasmon-fluorophore coupling and not easy scalability[9].

2D-lattice of gold nanoparticles (AuNPs), made by block copolymer micelle nanolithography (BCMN)[33], provide an easy way to overcome most of the aforementioned challenges. The affordable and scalable fabrication and the facile tunability of their plasmonic properties are the main strengths of these devices[33]. The plasmonic behavior of a 2D AuNP lattice is closely related to the ratio $R$ between the nanoparticle diameter $D$ and the interparticle distance $d$. When a nanoparticle is placed in close proximity to its nearest neighbors ($R > 2/3$)[34], interaction among the localized surface plasmons (LSPs) gives rise to long-range collective oscillation[35,36]. Such a collective effect is negligible if $R < 2/3$, in which case the plasmonic response of the 2D-lattice is well-described by a system of decoupled LSPs[34].

A key issue to address when such plasmonic substrates are used in biosensing is the biofunctionalization of the active surface since it deeply affects both the sensitivity and specificity[37]. Antibodies (Abs) are the prime candidates as bioreceptors thanks to their inherent specificity, versatility, and reliability. However, due to their moderate long-time stability and the need to immobilize them both with the right orientation and high surface density, the accomplishment of a robust and effective Ab surface functionalization is still an open issue[38]. In this respect, the well-established photochemical immobilization technique (PIT) is a simple, fast and effective strategy to tether Abs onto gold surfaces with one fragment antigen-binding (Fab) exposed to the surrounding environment[39]. Besides antibodies, aptamers are nucleic acids which have been evolved to bind to targets through molecular evolution approaches as alternative class of bioreceptors. Combining aptamers and antibodies has been proven as a powerful approach for molecular recognition[40], but such approaches have not been investigated for PEF-based sensors.

In the present work, we describe an ultrasensitive and cost-effective PEF-based immunosensor—consisting of 2D AuNP array functionalized by PIT—able to detect proteins at femtomolar level in whole human blood. Specifically, we measured the concentration of Plasmodium falciparum lactate dehydrogenase (PfLDH), a malaria biomarker present at nanomolar level in red blood cells but only at picomolar levels in serum of infected people[41]. Malaria is still one of the main causes of disease-related deaths worldwide[42,43] and it is caused by Plasmodium parasites such as P. vivax and P. falciparum, the latter accounting for 90% of mortality worldwide[44]. The conventional serologic antibody-based rapid diagnostic tests rely upon the PfLDH detection in pretreated blood and offer rapid and cost-effective malaria diagnosis. However, the poor LOD as well as the transportation and storage difficulties of pre-functionalized devices in tropical environment make such tests unfit for early diagnosis and screening[45–47]. The ultrasensitive PEF-based device in combination with a unique functionalization procedure (PIT), which can be accomplished in a few minutes, results in an immunosensor suitable for detecting PfLDH at femtomolar level in the whole blood without any pretreatment and preconcentration steps. The fluoroimmunoassay proposed here is inherently specific since the recognition of PfLDH takes place in a sandwich scheme by Abs from the bottom and aptamers from the top; thus, it can be immediately extended to other analytes by replacing the bioreceptor layers.

## Results

**Fluorescent aptamer-based immunoassay.** BCMN was employed to produce arrays of ordered AuNPs with tunable density, size, and interparticle distance[33]. Diblock copolymers with amphiphilic character were dispersed in a non-polar solvent (toluene) obtaining reverse micelles with a spherical shape, a hydrophilic core, and an outer hydrophobic shell (Fig. 1a). The resulting micelle can house the gold precursor inside, allowing the formation of AuNPs covered by a hydrophobic shell (PS-AuNPs). The substrates were then dipped into a solution containing PS-AuNPs by a dip coater to ensure a careful tuning of the dipping speed. The PS-AuNPs were transferred on the non-polar glass surface by hydrophobic interaction giving rise to a self-assembled

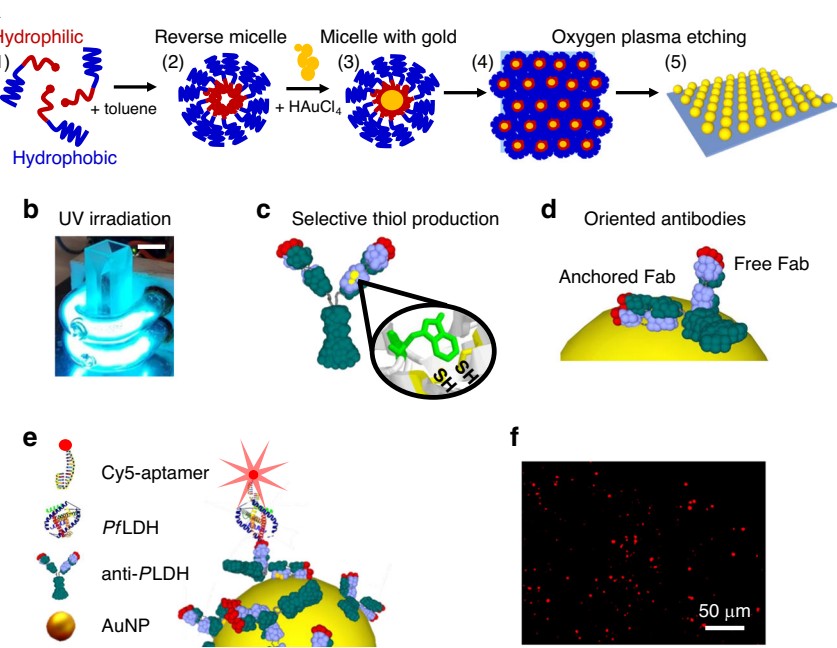

**Fig. 1 Operating principle of the device. a** Fabrication process of AuNP array by BCMN: (1) dispersion of diblock copolymers with amphiphilic character in toluene solution; (2) formation of reverse micelles with hydrophilic core and outer hydrophobic shell; (3) loading of the gold precursor inside the micelles; (4) sticking of the PS-AuNPs on the substrate through hydrophobic interaction; (5) immobilization of the AuNPs on the substrate after copolymer etching. **b** Low pressure mercury U-shaped UV lamps used to carry out the biofunctionalization of AuNPs with antibodies through PIT. A standard 10 mm cuvette can be easily housed inside the internal volume (the length of the scale bar in the top-right corner is 1 cm). Given the proximity of the cuvette to the lamps and the wrapping geometry, we estimated that the solution was exposed to an UV-irradiation of 0.3 W/cm². **c** UV irradiation of the Abs leads to the production of four thiol groups (two of them are not visible in the figure). **d** The position of the thiols, opposite with respect to the plane containing the antibody Fabs, allows to immobilize the Abs with one of their binding sites exposed to the surrounding environment. **e** Sketch of the Ab-*Pf*LDH-aptamer sandwich scheme used throughout the experiments. The adoption of Abs as bottom receptor layer and aptamers as top layer allows the fluorophore to be placed at a distance of approximately 10 nm from the surface. **f** Example of fluorescence picture acquired at 1 pM (35 pg/mL) *Pf*LDH concentration. The red spots arise from the fluorescence emitted by a single Cy5 molecule.

close-packing honeycomb arrangement. Then, the copolymers were etched by oxygen plasma treatment leaving the AuNPs immobilized on the glass surface at predetermined positions (Fig. 1a). Subsequent incubation in a gold solution entailed the growth of the AuNP size[48]. The resulting reduction of interparticle distance brought about an increase of the ratio $R$ up to a value of 2.5 that is large enough to activate collective plasmonic effects of the array[34,36].

Gold NPs on the surface were functionalized with pan malaria antibodies (anti-*P*LDH) using PIT[39,49,50]. Such a technique consists of UV irradiation of Abs (Fig. 1b) that leads the selective photoreduction of the disulfide bridge in specific cysteine–cysteine/tryptophan (Cys–Cys/Trp) triads[51]. The breakage of such Cys–Cys bonds in both Ab Fab fragments produces four free thiol groups (Fig. 1c), two of which are able to interact with the proximal gold surface giving rise to a covalent Ab tether. Functionalization by PIT assures control over the orientation of the immobilized Abs, with one of their binding sites exposed to the surrounding environment[39] (Fig. 1d). Moreover, the technique leads to a close-packing arrangement of the Abs[52,53], demonstrated by the plasmon resonance of the device. In experiments where we varied the concentration of the anti-*P*LDH over large intervals (see Supplementary Fig. 1), the wavelength shift ($\Delta\lambda_p$) showed no significant variations for concentrations larger than 50 μg/mL, that is a threshold above which no more free AuNP surface is available for Abs functionalization. We therefore used the value of concentration 50 μg/mL throughout the work. The close-packed biofunctionalization is also evident by the lack of $\Delta\lambda_p$ changes after the

blocking step (Supplementary Fig. 2). We estimated the value of Ab density (number of Abs per AuNP) as ~50, which corresponds to the maximum number of antibodies that can be anchored onto a sphere of diameter 50 nm (the steric hindrance of a single immobilized Ab is ~150 nm²)[52].

Figure 1e shows the sandwich scheme used for the immunosensor. The bottom bioreceptor layer (anti-*P*LDH) provides a reliable and effective targeting of any malaria biomarkers *Plasmodium* lactate dehydrogenase (*P*LDH)[54], whereas the top bioreceptor layer (aptamers) provides a cheap and versatile labeling of the analyte with fluorescent tag and ensures an extremely high specificity against *Pf*LDH[55]. Aptamers in the top layer enable the fluorophore to be placed at an optimal distance from the AuNP of ~10 nm, assuring optimal PEF amplification[14,15]. The fluorescence signal was recorded by a fluorescence microscope and the resulting pictures were analyzed by an image processing software to retrieve the corresponding intensity. Figure 1f shows a typically processed image collected at low *Pf*LDH concentration, in which the original non-flat background was removed by using the "rolling ball" algorithm[56] (see Supplementary Fig. 3) and the resulting red spots arose from photons emitted by a single fluorophore.

**Plasmonic response of a honeycomb lattice.** We used a numerical finite difference scheme implemented in Lumerical to simulate the electromagnetic (EM)-field scattered by the gold spheres on the sensor surface. The "FDTD solutions" tool of Lumerical software works out the numerical solutions of the Maxwell's equations via the finite-difference time-domain

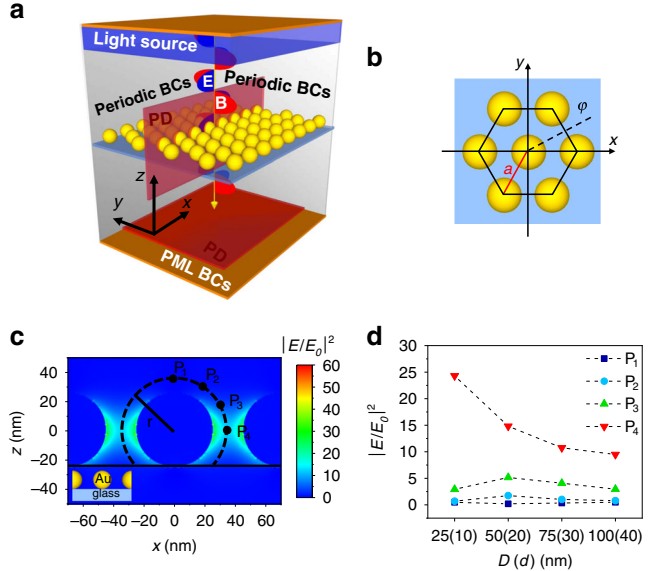

**Fig. 2 Optical properties of 2D AuNP lattice. a** Schematic representation of the simulation workspace consisting of plane wave source, plasmonic nanostructure, dielectric substrate (SiO$_2$ glass), photodetectors and appropriate BCs. Linearly $x$-polarized EM wave propagates through the 2D AuNP lattice. The transmitted photons are collected by a PD on the opposite side of the workspace, whereas the E-field intensity distribution is worked out by placing several PDs in correspondence of the lattice (not all depicted in the sketch). **b** Top view sketch of the simulation unit cell; the centers of the spherical AuNPs belong to the plane $z = 0$. The parameters $a$ and $\varphi$ are the lattice constant and the azimuthal angle, respectively. **c** Example of E-field distribution normalized to the incident radiation worked out in the plane $y = 0$ for $x$-polarized EM plane wave incident on a lattice made of AuNPs with diameter 50 nm and interparticle distance 20 nm. The points P$_1$–P$_4$ are at a distance of 10 nm from the nanoparticle surface at polar angle $\theta = 0°$, 30°, 60°, and 90°, respectively. **d** Gain of the E-field intensity as a function of AuNP diameter (interparticle distance).

(FDTD) method within a Mie problem-like workspace. A schematic representation of the simulation workspace and unit cell are shown in Fig. 2a, b, respectively. The electric field **E** (E-field) intensity exhibits a maximum in the $x$–$y$ plane along the polarization direction, while it shows a minimum in the transverse direction (see Supplementary Fig. 4).

By rotating the direction of polarization around $z$, we observed an azimuthal E-field intensity modulation with an expected period of 60°. Since all the polarization directions have to be taken into account when non-polarized light is used (e.g., light emitted by a fluorophore), the actual local E-field intensity is the result of the average of all such contributions (see Supplementary Fig. 5a). It turns out that the gain factor $|E/E_0|^2$ slightly modulates around a value of 4 when $\varphi$ is changed from 0° to 360° and the distance is 10 nm from the nanoparticle surface (see Supplementary Fig. 5b), thereby suggesting that the E-field intensity experienced by the fluorophore has a relatively weak azimuthal dependence.

The analysis as a function of the polar angle $\theta$ is reported in Fig. 2c that shows the distribution of the E-field intensity in the $y = 0$ plane, resulting from the interaction of an $x$-polarized plane wave with a 2D honeycomb lattice (Fig. 2b) made of 50 nm AuNPs distanced 20 nm one from the other. Due to the plasmon coupling, the largest values of the field intensity are recorded along the direction of polarization ($x$) while the smallest values are measured in the transverse direction to the plane $z = 0$. In

particular, the gain is more effective in the interval $45° \leq \theta \leq 135°$ that contains approximately 70% of the available surface.

Aiming at optimizing the performance of the substrate, we conducted a test-campaign where the response of the system was analyzed as a function of the characteristics of the 2D lattice. In the simulations, we varied the particle size and interparticle distance over large intervals (25 nm/10 nm, 50 nm/20 nm, 75 nm/ 30 nm, 100 nm/40 nm) keeping constant their ratio $R = 2.5$, a value that is large enough to warrant a high enhancement factor[34,36]. Figure 2d shows the enhancement of the E-field intensity for several polar angles ($\theta = 0°$, 30°, 60°, 90°) at a distance of 10 nm from the AuNP surface (points P$_1$–P$_4$ in Fig. 2c). Despite the strongest gain was achieved for the smallest particles, the need to allow the formation of the Ab-$Pf$LDH-aptamer sandwiches in the interparticle gaps suggested us to consider 50 nm/20 nm as the optimal choice to guarantee good enhancement of the E-field intensity while assuring that the process of functionalization and detection occurs correctly.

**Optical and morphological characterization of the 2D AuNP array.** The characterization of the substrate was performed by scanning electron microscopy (SEM) and UV–Vis spectroscopy (Fig. 3). SEM images of the device (Fig. 3a, b) show the fabrication process capability to attain maximum control over the geometrical characteristics of the device, including particle size and particle distance. Aiming at realizing a substrate with a unique collective plasmonic behavior, the nanoparticle growth is carried out to increase the $R$ value (Fig. 3c). The nanoparticle diameter increased approximately five-fold while the interparticle distance reduced three-fold by holding the lattice period equal to ~70 nm. The $R$ value went from 0.17 to 2.5 warranting a collective response of the AuNPs immobilized on the substrate[34,36]. The size distribution of the AuNPs before the gold growth process (blue columns) is peaked at approximately 10.4 nm with a full width at half maximum (FWHM) of 1.4 nm, while that after nanoparticle growth (red columns) has a mean of ~48 nm and a FWHM of 6 nm (Fig. 3d). The smaller peak at approximately 61 nm (red columns) is due to fewer AuNP clusters as a byproduct of the gold nanoparticle growth process. The center-to-center distance $d_{C–C}$, measured as the distance of each AuNP centroid with its nearest neighbors follows the distributions in Fig. 3e. The mean values $\bar{d}_{C–C}$ are 69 nm (blue columns) and 68 nm (red columns) with standard deviations of 8 nm and 14 nm, respectively. The high similarity of such distributions confirms the holding of most of AuNP positions also after the growth process, whereas the relatively large values of standard deviation for $\bar{d}_{C–C}$ can be ascribed to defects, such as clusters and vacancies. The occurrence of $d_{C–C}$ lower than $D$ after the growing process (red histograms in Fig. 3d, e) is due to the lack of the AuNP spherical shape arising from nanoparticle clustering (Fig. 3b).

The 2D AuNP array was also characterized optically by measuring its extinction spectrum whose resonance wavelength and shape are not only strongly dependent on the ratio $R$[34,57], but also on the regularity holding at high macroscopic level, a feature warranted by the array fabrication procedure adopted in this work (see Supplementary Fig. 6 for SEM pictures on large scale). Figure 3f shows a plasmonic resonance (green continuous line) occurring at approximately 650 nm whose width is large enough to contain both the excitation (blue dotted curve) and emission (red dashed curve) spectra of cyanine 5 (Cy5), fluorophore used in this experiment, which has an excitation peak at 649 nm, an emission peak at 666 nm and a quantum yield (QY) of 0.27[58]. The superposition of the plasmonic resonance to both excitation and emission peaks of the fluorophore provides an ideal condition for achieving a PEF amplification since both FRET and Purcell effect

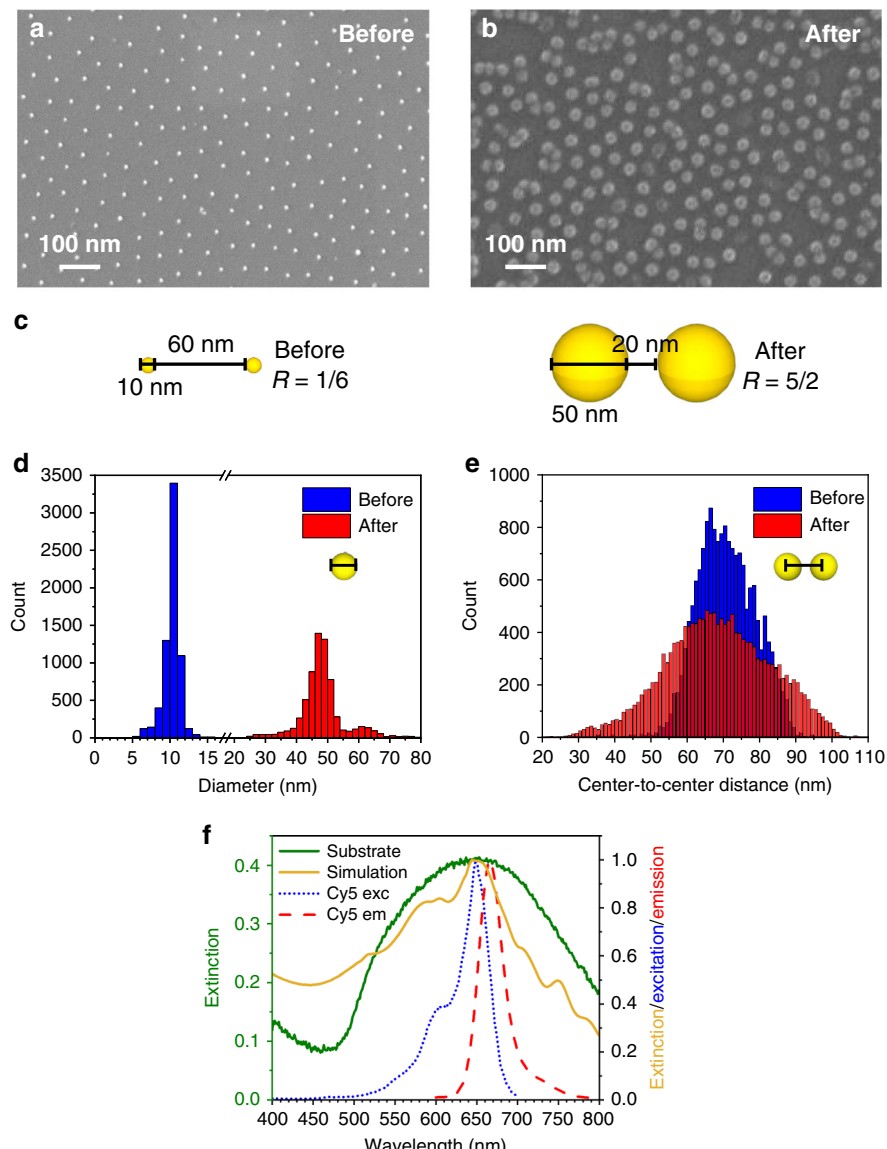

**Fig. 3 Substrate characterization. a, b** Top view SEM images of the AuNP array show high regularity of nanoparticle shape and size. Defects arising during the AuNP growth step, such as clusters and holes are randomly distributed on the substrate. **c** Sketch of the AuNP growth process. The nanoparticle diameter increases approximately five-fold while the interparticle distance reduces three-fold by holding the lattice period (center-to-center distance among nearest neighbors) equal to ~70 nm. The *R* value goes from ~0.17 to ~2.5 warranting a collective plasmonic behavior of the AuNPs immobilized on the substrate. **d** Histograms of nanoparticle diameter before (blue columns) and after (red columns) incubation with gold growth solution. The distributions are peaked at ~11 nm and ~48 nm, respectively. **e** Center-to-center distance histograms exhibit similar distributions confirming the holding of most of AuNP positions also after the gold growth process. The wider distribution delivered by the substrates after the AuNP growth (red columns) is due to the presence of lattice irregularities and defects arising during the incubation process. **f** The experimental extinction spectrum of the substrate (green continuous line) is well reproduced by simulating the optical response provided by the actual substrate morphology as measured by SEM (gold continuous line). The excitation and emission peaks of Cy5 (dotted blue line and dashed red line, respectively) are encompassed within the substrate plasmon resonance.

can take place[11,13,14]. Such a superposition can be easily realized at essentially any substrate plasmon resonance given the wide variety of fluorescent dyes apt to be used as aptamer label. The experimental extinction spectrum is well reproduced by that one worked out by considering the real morphology of the substrate as provided by SEM (gold continuous line), the latter being obtained by averaging the extinction spectra resulting from fifteen regions of interest sampled from the SEM image shown in Supplementary Fig. 7a (an example of the rendering provided by Lumerical is shown in Supplementary Fig. 7b, whereas some of the simulated extinction spectra are reported in Supplementary Fig. 7c). The consistent agreement between the experimental extinction spectrum and that worked out by FDTD provides

assurance that the observed resonance profile is the result of the collective plasmonic behavior exhibited by a 2D array made of spherical AuNPs, and that the quite large experimental spectral broadening also arises from the partial reduction of the degree of the spatial order due to the growth process.

**Fluorescence-based immunoassay performance.** The hybrid sandwich scheme shown in Fig. 1e was used in the 2D AuNP array functionalized by PIT for detecting *Pf*LDH at ultra-low concentrations in one of the most complex matrices as it is the whole human blood. Two preliminary binding kinetic studies were performed in order to find out the incubation time necessary

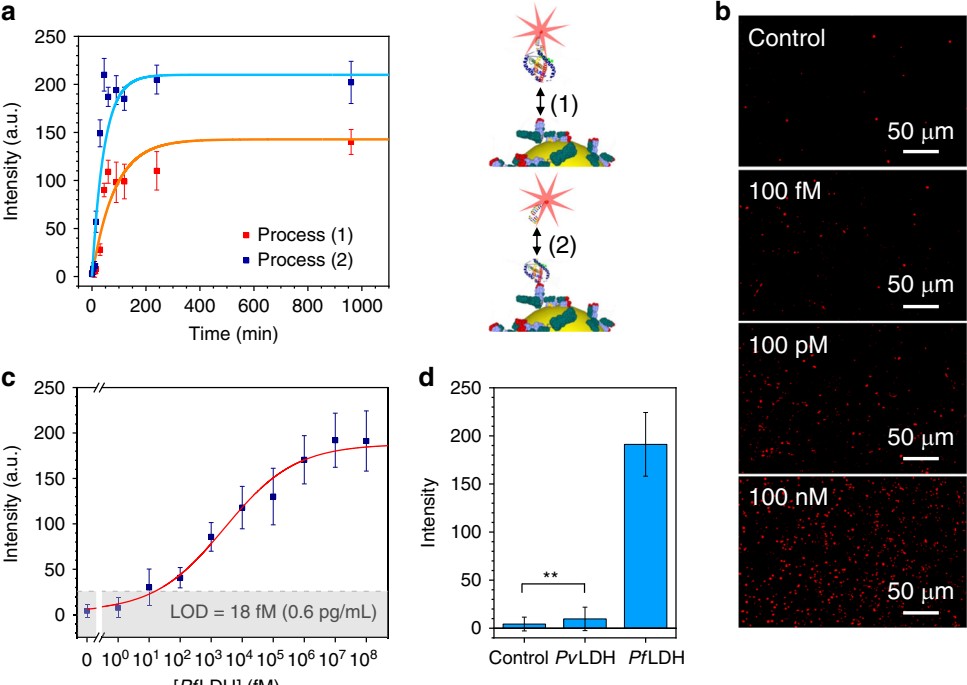

**Fig. 4 Fluorescence-based immunoassay. a** Kinetic curves related to (1) Ab-analyte and (2) analyte-aptamer binding dynamics. Both processes reach their equilibrium in a short time by incubation and tilting mixing (80 min and 45 min, respectively). The time difference is ascribable to the significantly larger *Pf*LDH mass. The data are well fitted by exponential curves (orange and blue continuous lines). **b** Fluorescence images acquired at different *Pf*LDH concentrations in spiked human blood. **c** Calibration curve (fluorescence intensity vs *Pf*LDH concentration in spiked human blood) of the immunoassay for *Pf*LDH concentration in the range 1 fM to 100 nM (35 fg/mL to 3.5 μg/mL). The data are best fitted by the four-parameter Hill equation (red solid line). The gray region represents the $3\sigma$ noise level recorded in uncontaminated human blood (LOD = 18 fM (0.6 pg/mL)). **d** Specificity of the immunoassay against *Pv*LDH (90% residue identity with *Pf*LDH) at a concentration of 100 nM in spiked human blood. No cross-reaction detected with the main *Pf*LDH competitor highlights the extremely high specificity of the aptamers used as top bioreceptor layer (**$p$-value < 0.001). All the data underlying averaged value are presented as mean value ± SD and are representative of ten technical repeats. Source data are provided as a Source data file.

to reach the dynamic equilibrium for both (1) Ab-analyte and (2) analyte-aptamer binding. In order to detect the occurrence of the process (1), the analytes were made visible by complexing them with the fluorescently labeled aptamer by assuming that the small size of the aptamers did not significantly change the process kinetics, whereas as it concerns the process (2), fluorescent aptamers were incubated with Ab-analyte complexes immobilized on the substrate. Both the processes (1) and (2) show very similar kinetics (Fig. 4a and Supplementary Fig. 8 for the corresponding fluorescence images) that are well fitted by exponential curves with a time constant of 80 ± 30 min and 45 ± 15 min, respectively (orange and light blue continuous line in Fig. 4a). Since the fluorescence measurement could be carried out by commercial fluorescence readers, which are able to deliver multi-response within few minutes, the estimated detection time for the whole analysis is shorter than 3 h. The slight difference in the fluorescence intensity at the dynamic equilibrium between the two kinetics can be ascribed to the pre-incubation among the free analytes and the fluorescent aptamers in the process (1), which leads to a less effective analyte binding since the aptamers might have targeted the *Pf*LDH proteins on both sides, hindering their capture by Abs[55].

Aiming at analyzing the whole blood, we had to dilute it to avoid clotting and turbidity of the solution to be incubated. The *Pf*LDH was spiked into uninfected blood that was 1:100 diluted in 1 mL of 25 mM Tris buffer achieving a good trade-off between high signal and treatable solution. Rather than exploiting additive concentrations, each substrate was used for a single concentration measurement so to inherently test the reliability and reproducibility of the detection procedure. Some of the resulting

fluorescence images are shown in Fig. 4b (see Supplementary Fig. 9 for more images), from which one can observe that the number of the fluorescence spots is significantly higher than that the control down to a *Pf*LDH concentration of 100 fM (3.5 pg/mL). The intensity of a single spot turns out to be highly correlated to its area *A* (see Supplementary Fig. 10) with an average area of 15 μm² corresponding to ~36 pixels (pixel area = 650 × 650 nm²).

The FE factor is defined as $G = I_{PEF}/I_0$, where $I_{PEF}$ and $I_0$ are the fluorescence intensity from a single fluorophore with and without the plasmonic substrate, respectively. The latter measurement was realized by drop-casting 100 μL of a solution containing 2.5 nM of Cy5-aptamer complex on a bare microscope slide. The dried drop exhibited a quite circular shape with a diameter of ~12 mm, whereas the fluorescence imaging showed an irregular "coffee ring" whose thickness never exceeded 0.1 mm. In order to assess the whole fluorescence intensity, the drop area was sampled in regions of interest, finding out that the intensity in the ring was 4-fold higher than that of the inner region (see Supplementary Fig. 11). To determine the upper limit for the fluorescence intensity, we considered the emission area as given by a circle with 12 mm diameter and an outer annulus 0.1 mm thick obtaining a value $F_0 = k_{ins} N_0 I_0$, with $N_0$ the number of fluorophores in the drop ($1.5 \times 10^{11}$ Cy5-aptamer complexes) and $k_{ins}$ an instrumental constant that includes the intensity of the incident radiation. In contrast with the previous case, the fluorescence with the plasmonic substrate consists of separated emitters (Fig. 4b); thus, if $N_{PEF}$ is the number of spots, we have $F_{PEF} = k_{ins} N_{PEF} I_{PEF}$. By measuring $F_{PEF}$ with our 2D AuNP array, we obtained the lower limit $G = 7 \times 10^4$ for the FE factor.

Although the physics underlying the fluorescence enhancement by plasmonic nanostructures is still to be fully understood[9,59], we can safely assess that such a high value of $G$ is also the result of the overlap of the plasmon resonance with both excitation and emission peaks[13,14,17].

The fluorescence intensity $F$ as a function of the $Pf$LDH concentration is reported in Fig. 4c. The experimental data are well fitted by the four-parameter Hill equation

$$F([Pf\text{LDH}]) = F_1 + \frac{F_2 - F_1}{1 + \left(\frac{K}{[Pf\text{LDH}]}\right)^n} \tag{1}$$

with $F_1 = 4 \pm 2$ arb. units, $F_2 = 190 \pm 10$ arb. units, $K = (2.7 \pm 1.4) \times 10^3$ fM, the Hill coefficient $n = 0.38 \pm 0.08$ and $\chi^2 = 2$. The dynamic range extends over five decades (from 10 fM to 1 nM), whereas a LOD of 18 fM can be estimated by considering the signal threshold as three standard deviations of the control value (gray region in Fig. 4c).

The specificity of the biosensor is very challenging when the analyte has to be detected in a complex matrix such as human blood, but it is also particularly important for malaria diagnostics in areas where both *P. falciparum* and *P. vivax* are endemic as the clinical decision-making for drug treatment is different for infection with each species. In this respect, we studied the immunoassay response against the *Plasmodium vivax* lactate dehydrogenase (*Pv*LDH), which has a high functional and sequence similarity with *Pf*LDH (90% residue identity[60]). The *Pv*LDH was spiked into uninfected blood that was 1:100 diluted in buffer solution and the resulting fluorescence intensity is comparable to that measured from the control (Fig. 4d). Although the bottom anti-*P*LDH layer in our sandwich scheme captured any *P*LDH malaria biomarkers, no significant cross-reaction was detected in the case of *Pv*LDH because of the extremely high specificity of the aptamers used as top bioreceptor layer. This is consistent with reports that the aptamer used in this assay is highly specific to *Pf*LDH over *Pv*LDH[61].

## Discussion

The gold standard for diagnosing malaria relies on the microscopic examination of blood films that not only necessitates trained personnel, but is unsuitable for a rapid diagnosis, which is required for a favorable prognosis of the disease[62]. Thus, big efforts are put in trying to develop rapid test based on a technology accessible even to the populations more exposed to the risk of infection. The most effective strategy largely adopted entails the detection of some of the enzymes expressed by the *Plasmodium* parasites that cause the disease. Lactate dehydrogenase is expressed in all *Plasmodium* species whilst an alternative biomarker, histidine rich protein II (HRP2), is only expressed in *P. falciparum*. Typically, for *P. falciparum* detection, LDH is preferred to HRP since the former better correlates with the parasite density[41], however often species-specific detection requires tests against both LDH and HRP. This work describes an immunosensor capable to specifically detect *Pf*LDH in whole blood at femtomolar level (LOD < 1 pg/mL), which is a limit of detection several orders of magnitude lower than the rapid diagnostic tests[63] or the commercial ELISA kits[41,64]. Such a LOD is reached thanks to a unique optimization of two ingredients of a biosensor: the transduction mechanism and the surface biofunctionalization, coupled with the recognition of the biomarker by both antibodies and aptamers. Furthermore, the functionalization procedure of PIT can be carried out by a simple UV lamp in a few minutes, thus we anticipate that such an approach can be carried out in low resource settings which would be critical for many malaria diagnostic applications. Therefore, we foresee that our approach may provide a solution to the cold transport challenges common to the presently used immunochromatographic lateral flow assays[45–47].

Whilst the *Pf*LDH and *Pv*LDH investigated for the data herein were expressed in *E. coli*, previously solved crystal structures of these proteins in complex with aptamers show that these proteins do reflect the native conformation of these proteins[55,65]. It has been highlighted that sensitivity is a critical issue for malaria rapid diagnostic tests. The limit of detection for *P*LDH-based immunochromatographic rapid diagnostic tests are typically in the range of 25 ng/mL[63]. The approach we have developed herein is over 1000-fold more sensitive. This opens up the possibility of point-of-care detection of *P*LDH in saliva which could transform malaria diagnosis as then one could diagnose non-invasively without the need for drawing blood from a patient. Such an approach would also reduce the risk of transmitting blood-borne pathogens. *P*LDH has been demonstrated to be present in saliva[66], but there is no technology yet developed with sufficient sensitivity. Attempts to diagnose in saliva using conventional approaches lead to a high number of false negatives[46]. The ultrasensitive approach we describe herein may provide a cost-effective solution providing the first non-invasive approach for a point-of-care diagnosis of malaria. A clinical study will be required to address this possibility.

Regarding transduction, we exploited the enhancement of the fluorescence intensity occurring by means of plasmonic nanostructures. Such a process is critically dependent on a number of conditions, the most noticeable being (i) the geometry of the nanostructure, (ii) the amount of overlap between the plasmonic resonance with both the fluorophore absorption and emission and (iii) the distance between the fluorophore and the surface. The constraint of realizing a device for practical applications suggested us to adopt the BCMN, which allows one to rapidly fabricate ordered arrays of gold nanoparticles over large surfaces[33], readily usable as a substrate for multiwell plates. In order to choose optimal values for the diameter $D$ and the inter-particle distance $d$, we fixed their ratio as $D/d = 2.5$—so to have significant interaction among the gold nanoparticle—and examined how the electric field intensity distribution varies as a function of nanoparticle size. Numerical simulations showed that the maximum amplification of the incident field is achieved at $D = 50$ nm and $d = 20$ nm, a configuration hardly realizable by conventional techniques.

Since the nanoparticles are realized by a growing process, their interparticle distribution has a standard deviation of 14 nm, whereas their surface shows some degree of roughness. Rather than being a drawback, such features lead to a relatively wide plasmon resonance peak that realizes the so-called dual enhancement mechanism[19], in which the plasmon resonance overlaps with both the excitation and emission spectra of the fluorophore (point ii). The fluorescence enhancement only occurs when the fluorophore is placed at approximately 10 nm from the surface (point iii)[14,16]. To achieve precise positioning of the fluorophore, we exploited an effective surface biofunctionalization procedure (PIT), which binds the Abs so that their Fab region is at ~5 nm from the surface[39,67], with a sandwich scheme that included a tagged aptamer on the top. Since the size of both the *Pf*LDH and the aptamer is only a few nanometers, the fluorophore automatically positions itself at the optimal distance from the surface, and this was demonstrated by the measured-experimental enhancement factor of $7 \times 10^4$ of the system.

As previously shown, beside oriented antibodies, the surface functionalization adopted in this work leads also to their close packed distribution both on AuNPs[52,53] and flat surfaces[68]. In this way, not only the interacting area of the biosensor is fully exploited thereby obtaining high sensitivity and low LOD, but also the specificity is inherently warranted by the presence of the antibodies (and even more increased in our scheme by the

presence of the aptamers). As a result of the occurrence of optimal conditions for detecting *Pf*LDH, we were able to measure a LOD of 18 fM (0.6 pg/mL), which is comparable to that recently measured by means of a much more complex Luminex technology[69].

By building on our results, two straightforward perspectives can be explored and even combined to make the clinical potential of our approach of significant impact: multiplexed and ultra-sensitive (at attomolar level) detection. The former can be accomplished by functionalizing each section of a large substrate (e.g., multiwell plate) with different antibodies so that several analytes can be easily distinguished by their position on the substrate. On the other hand, detection at the attomolar level can be achieved by detecting the analyte in serum or plasma since these matrices would not require the 1:100 dilution carried out in this work to reduce the turbidity of the whole blood.

## Methods

**Materials and chemicals**. Diblock copolymers (P18226-S2VP) were purchased from Polymer Source Inc. (Dorval, Canada) and were made by polystyrene($x$)-b-2-poly-vinylpyridine($y$) (PS($x$)-b-P2VP($y$)), in which $x = 30,000$ g/mol and $y = 8500$ g/mol are the molecular weight of polystyrene (PS) and poly(2-vinylpyridine) (P2VP), respectively. Toluene (99.8%), gold(III) chloride trihydrate (HAuCl$_4$·3H$_2$0), silver nitrate (AgNO$_3$) and ascorbic acid were purchased from Sigma-Aldrich; acetone (≥99.0%), 2-propanol (≥99.5%) and ethanol (≥99.5%) were purchased from Merck Millipore; hexadecyltrimethylammonium bromide (CTAB) (≥99.0%) was purchased from Fluka; bovine serum albumin (BSA) (fraction V IgG free, fatty acid poor) was obtained from Gibco. Ultrapure deionized water used for all aqueous solutions was dispensed by Milli-Q® system (18.2 MΩ cm resistivity). 10 mM phosphate-buffered saline (PBS) (NaCl 10 mM, NaH$_2$PO$_4$ 10 mM, Na$_2$HPO$_4$ 10 mM, MgCl$_2$ 1 mM, pH 7.1) and 25 mM Tris-HCl buffer (NaCl 100 mM, imidazole 20 mM, Tris 25 mM, HCl 25 mM, pH 7.5) were prepared by dissolving the reagents (purchased from Sigma-Aldrich) in ultrapure water. Pan malaria antibody (monoclonal anti-*P*LDH antibody clone 19g7) was produced by Vista Laboratory Services (Langley, USA). The recombinant *Plasmodium falciparum* lactate dehydrogenase (*Pf*LDH) and *Plasmodium vivax* lactate dehydrogenase (*Pv*LDH) were obtained from bacterial expression as described previously[61]. The malaria 2008s aptamer labeled with cyanine 5 tag (5′-Cy5-CTG GGC GGT AGA ACC ATA GTG ACC CAG CCG TCT AC-3′) was synthesized by Friz Biochem GmbH (Neuried, Germany). Millex® syringe filters (pore size 0.20 μm) with hydrophilic polytetrafluoroethylene membrane were purchased from Merck Millipore; Superslip™ coverslips (borosilicate glass, thickness 0.13–0.17 mm) were purchased from Thermo Fisher Scientific and cut by diamond tipped glass cutter.

**Fabrication of 2D AuNP array**. 29.2 mg of diblock copolymer P18226-S2VP was added into 15 mL of toluene under vigorous stirring and controlled conditions (argon inert gas, O$_2$ < 1 ppm, H$_2$O < 0.1 ppm) for 72 h achieving homogeneously dispersed reverse micelle. Then, 15.7 mg of HAuCl$_4$·3H$_2$0 was loaded in the solution by holding the vigorous stirring for 72 h allowing the gold precursor to be housed into the hydrophilic core of the micelles. When the gold powder was completely dispersed, the yellowish solution was filtered to remove micelle aggregates and impurities. The solution can be stored by holding the vigorous stirring up to approximately six months. Before depositing the PS-AuNPs on the glass coverslips (10 × 8 mm$^2$), the substrates were cleaned by sonication for 5 min in acetone, 2-propanol and pure ethanol sequentially, and dipped in toluene in order to make non-polar the surface so that the hydrophobic shells could stick on that. Then, the glass coverslips were dipped into the PS-AuNPs solution by dip-coater to carefully tune the dipping speed. A dipping speed of 0.6 mm/s provided an optimal coverage of the glass surface in terms of both PS-AuNPs close-packing and long-range regularity (see Supplementary Fig. 6). The copolymers were etched by oxygen plasma treatment (0.8 mbar pressure, 200 W power, 30 min) leaving the AuNPs immobilized on the glass surface at prefixed positions. Afterward, the substrates were incubated with 2 mL gold growth solution (CTAB 190 mM, HAuCl$_4$·3H$_2$O 42 mM, AgNO$_3$ 8 mM, ascorbic acid 100 mM) for 2 h in dark condition. Then, the substrates were copiously rinsing by ultrapure water and stored in dark condition until use.

**Substrate biofunctionalization and blocking**. The functionalization of AuNPs with anti-*P*LDH was achieved by PIT. 1 mL aqueous solution of anti-*P*LDH (50 μg/mL) was irradiated by an UV-lamp (Trylight®, Promete S.r.l.) for 30 s and then flowed onto the substrate. The UV source consisted of two U-shaped low-pressure mercury lamps (6 W at 254 nm) in which a standard quartz cuvette could be easily housed (Fig. 1b). By considering the wrapping geometry of the lamps and the proximity of the cuvette, the irradiation intensity used for the thiol group production was approximately 0.3 W/cm$^2$. Such an intensity is low enough to avoid

any significant photolysis of the disulfide bridge that poorly absorbs at 254 nm[70]. Then, the substrates were rinsed by ultrapure water to remove the unbound Abs. As it concerns the blocking of the free active surface, 1 mL of BSA aqueous solution (50 μg/mL) was used to prevent nonspecific adsorption. Afterward, the samples were copiously rinsed by ultrapure water and stored in PBS solution (10 mM) at room temperature until use.

**PfLDH capture by immobilized Abs**. The desired amount of *Pf*LDH was spiked into 1 mL of uninfected whole blood (1:100 diluted in 25 mM Tris buffer). The functionalized substrates were incubated with 1 mL of contaminated blood solution for 2 h at room temperature; a tilting laboratory shaker was used to improve the analyte diffusion. Then, the samples were copiously rinsed by Tris buffer (25 mM) and ultrapure water to remove the unbound proteins.

**PfLDH targeting with fluorescently labeled aptamers**. The samples were transferred to 1 mL of PBS solution (10 mM) containing 0.1 μM of malaria 2008s aptamers labeled with Cy5 tag. The solution was gently shaken for 2 h in dark condition by a tilting laboratory shaker so that the sandwich scheme reported in Fig. 1e is realized. Then, the samples were abundantly rinsed by PBS and ultrapure water to remove the unbound aptamers.

**SEM image analysis**. The substrate was observed by Zeiss LEO 1550VP field emission scanning electron microscope (FESEM) with a nominal resolution of 1 nm at 20 kV acceleration voltage. The recorded SEM images (Supplementary Fig. 12a shows an example of raw image at high magnification) were processed by ImageJ software to retrieve information about the substrate morphology. To this aim, each image was binarized to isolate the objects from the background (Supplementary Fig. 12b) and segmented by "Watershed" tool implemented in ImageJ to separate adjacent nanoparticles (Supplementary Fig. 12c). Then, object area $S$, perimeter $p$, shape descriptors (aspect ratio $AR$ and circularity $4\pi\, S/p^2$) and centroid coordinates were measured by "Analyze Particles" tool implemented in ImageJ. Supplementary Fig. 12d shows an example of processed SEM image, in which the objects are decomposed in outlines (black line) and inner area (orange filling). Given the round shape of the objects (Supplementary Fig. 13), nanoparticle diameter was estimated as $D = 2\sqrt{S/\pi}$, whereas the centre-to-centre distance distribution was carried out by calculating the distance of each centroid from its nearest neighbors.

**Processing and analysis of the fluorescence images**. Fluorescence images were recorded by Zeiss Axio Observer Z1 inverted phase contrast fluorescence microscope equipped by Zeiss Colibri.2 LED light source (module 625 nm), Zeiss Plan-Apochromat 10x/0.45 Ph1 M27 (FWD = 2.1 mm) objective, cube 50 Cy5 filter (excitation 625–655 nm/emission 665-715 nm) and pco.edge 5.5 sCMOS photodetector (scaling 0.650 μm × 0.650 μm per pixel, image size 2560 × 2160 pixels, scaled image size 1.66 mm × 1.40 mm, 16 bit dynamic range, 2 s exposure time was used for every image). The recorded fluorescence images were processed by ImageJ software. The "rolling ball" algorithm was used to remove the smooth continuous background from the images[56] (see Supplementary Fig. S3). The background was locally measured for each pixel by averaging over a ball around the pixel. Such a value is then subtracted from the original image flattening the spatial variations of the background. The rolling ball radius was set 10 pixels, a length reasonably higher than the size of the largest objects that are not part of the background. A threshold level slightly higher than the flattened background was set to segment the image whose whole intensity was measured by summing the signal contribution of every spots. To ease the readability of the calibration curve (Fig. 4c), the intensities were scaled down by an arbitrary factor. Aiming at carrying out a robust and reliable analysis of the fluorescence signal, ten images were randomly recorded for each sample and the mean of their intensity was found out.

**Simulations of the optical response**. The optical response of both the 2D AuNP lattice and the actual substrate was simulated by "FDTD solutions" tool implemented in Lumerical software. A linearly polarized EM radiation traveling along the $z$ direction was used to investigate the system. Photodetectors (PDs) conveniently positioned in the workspace could measure the intensity of the EM field over time. A PD was dedicated to measure the extinction spectrum of the nanostructure. Symmetric/anti-symmetric boundary conditions (BCs) set along $x$ and $y$ direction extend the plasmonic response over an infinite 2D array while reduce the simulation time by a factor 8 without worsening the accuracy of the results. Bloch BCs were used only for polarization study in order to compensate the phase shift arising when an EM disturbance with a non-zero angle should be re-injected at the opposite workspace site. Perfect matched layer BCs set in $z$ assures perfect absorption of the EM waves backscattered through the plane containing the light source and incident upon the opposite side of the workspace. The workspace was discretized over a mesh with a spatial resolution of 0.5 nm and 1.0 nm for the 2D AuNP lattice and the actual substrate, respectively. Such a choice assured high accuracy while keeping the simulation time within few hours. The AuNPs of the lattice were modeled as homogeneous gold spheres[71], while the bare substrate was represented as a thick dielectric layer of silicon dioxide (SiO$_2$)[72]. The extinction

spectrum of the actual substrate was carried out by importing the SEM morphology in Lumerical workspace. To this aim, the raw SEM image (Supplementary Fig. 14a shows an example at high magnification) was thresholded and binarized to create a mask template, in which the object pixel value was set equal to 1 and the background to 0 (Supplementary Fig. 14b). The centroid coordinates $(i_{nc}, j_{nc})$ of the $n^{th}$ object were retrieved by running "Analyze Particles" tool implemented in ImageJ software (Supplementary Fig. 14c). Following to the mask template, the centroid positions were used to shape spherical nanoparticles (diameter $D = 50$ nm) as

$$z(i_n, j_n) = \pm \sqrt{(D/2)^2 - (i_n - i_{nc})^2 - (j_n - j_{nc})^2}$$ (Supplementary Fig. 14d).

**Reporting summary**. Further information on research design is available in the Nature Research Reporting Summary linked to this article.

## Data availability
The data that support the findings of this study are available from the authors on reasonable request. Source data are provided with this paper.

## Code availability
The codes used to analyse the data are available from the authors on reasonable request.

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

## Acknowledgements

We would like to thank Ruoyan Wei for her help on substrate fabrication through block copolymer micelle nanolithography and Gabriela Figueroa Miranda for her advices on aptamer handling.

## Author contributions

A.M., B.DV., R.V., and D.M. conceived the project. A.M. carried out the experiments and collected the data under D.M. and A.O. supervision and administration. A.M., B.L., and F.G. worked out the numerical simulations. A.M., B.DV., R.V., and F.G. performed the data interpretation. J.A.T. and D.M. provided support on the use of aptamers. A.M., R.V., and J.A.T. wrote the paper. All authors were involved in the revisions.

## Funding

## Competing interests

The authors declare no competing interests.
