## [Peer Review File · Nature Communications]

Reviewer #1 (Remarks to the Author):

This is an outstanding manuscript and can be accepted in the present form. This concept of metal-enhanced fluorescence (MEF) assays have been known for almost 30 years. It was also known that both quenching and enhancement occurred. In almost all previous publications these effects were mentioned, but not quantified. The authors' careful testing of multiple substrates (particle sizes) allowed selection of conditions where the enhancement was greatest.

The substrate design is excellent. The use of reverse micelles, block polymers and subsequent growth of the gold nanoparticles can be performed over large areas at modest cost. The use of gold, rather than more commonly used silver, will allow the nanoparticles substrates to be stable for long periods of time.

There is one point which the authors can make more clear. What is the ??? between the 7×10^4 enhancement on page 10 and eq (1) on pH. The selection of $F_2 = 190$ is not clear.

Reviewer #2 (Remarks to the Author):

In this manuscript the authors report on a novel and very sensitive technique to measure malaria biomarkers. They make use of a plasmonic enhancement technique in order to significantly increase the fluorescence intensity, together with an aptamer which increases specificity to the target biomarker. The approach which they use is applicable to a wide range of different targets and so I believe that this paper would be of interest to a general audience in the field of biosensing. The authors describe the experimental methods in sufficient detail to permit reproducibility and make reference to appropriate previous work. They also provide extensive supplemental studies which support their main conclusions and which provide more details that will be of interest to readers. The level of sensitivity and specificity which is achieved is very impressive and is superior to currently used methods. Therefore, I believe that this manuscript should be published in Nature Communications.

I do however have a number of comments and questions that should be addressed by the authors:

1. Based on the electromagnetic analysis presented in figure 2(d) the authors choose a particle diameter of 50 nm and interparticle distance of 20 nm, claiming that will give the greatest fluorescence enhancement. However, in figure 3 (f) they show that the extinction spectrum of a substrate fabricated at those dimensions would lie outside the absorption and emission peak of the fluorescent dye. It is only due to fabrication imperfections that the extinction peak maximum is coincident with the absorption and emission peaks. Clearly, as demonstrated by their experimental work there is a fluorescent enhancement. However, they should provide a better explanation as to whether the 50-20 nm spacing is really optimal from a theoretical point of view, and whether the enhancement would still be observed if fabrication had been perfect. This would be important for other researchers seeking to follow up on this work.
2. It is assumed throughout (and stated on page 4) that the nanoparticles are hemispherical in shape. However, the highest magnification image (Fig 3(b)) does not provide enough detail to allow the reader to see whether that assumption is justified. The authors should provide a better analysis of the shape of the nanoparticles.
3. On Page 3, the authors point out that pre-functionalized biosensors for malaria often have transportation and storage difficulties. In the discussion section they should justify whether their approach reduces these difficulties.

4. On page 6 the authors appear to find it remarkable that their periodic structure would have an azimuthal E-field modulation with a period of 60 degrees. However given the hexagonal symmetry of the nanostructure that they simulate, this does not seem particularly remarkable to me.

5. On page 7, the authors state 'At all considered points, the enhancement decreases with the diameter of the AuNP, with the exception of P1 with an increment for $D=50\text{ nm}$ slightly larger than that for $D=25\text{ nm}$ '. What does that mean?

6. Figure 3 (d) and (e) shows the lattice spacing measured from the micrographs. The authors should provide more details as to how these spacings were calculated from the images.

7. Figure 4(b) captions refers to the sample as being 'contaminated' human blood. 'Spiked' is probably a more suitable word.

Reviewer #3 (Remarks to the Author):

The MS by Minopoli et al entitled "Ultrasensitive antibody-aptamer plasmonic biosensor for malaria biomarker detection in whole blood" describes the development of a plasmon-enhanced fluorescence immunosensor for the specific and ultrasensitive detection (femtomolar level) of Plasmodium falciparum lactate dehydrogenase (PfLDH) in whole blood.

The results are novel (a new immunosensor is described) and of interest to the malaria community but also to other fields working on the development of highly sensitive devices to detect microbial antigens. Given the nature of the work presented, no statistical analysis are included in the paper. I have no comments with regards to the development of the assay, as the procedures are out of my expertise. However, I have two main comments with regard to the validation and use cases of the assay developed.

Which is the expected use of an assay that allows detecting such low amounts of a Pf antigen? Are the authors considering its use for the validation of malaria rapid diagnostic tests (RDT)? For the use in the field? In the first sentence of the last paragraph in the discussion, authors seem to suggest its clinical potential. In which situations such a highly-sensitive test would be used? For what purpose? There have been several discussions in the malaria community about the use of a recently developed RDT which has higher sensitivity than standard RDTs, without achieving a consensus of the rationale for detecting extremely low parasite densities. Costs should be considered in this analysis.

Authors have used recombinant PfLDH spiked in blood to assess the performance of the assay. How was PvLDH produced? Also obtained from bacterial expression? How well did these proteins reflect the native confirmation of the antigens? Which is the LOD and the general performance of the assay when tested against PfLDH and PvLDH in clinical samples?

Response to Reviewer #1 (in *red italics* the reviewer's comment).

This is an outstanding manuscript and can be accepted in the present form. This concept of metal-enhanced fluorescence (MEF) assays have been known for almost 30 years. It was also known that both quenching and enhancement occurred. In almost all previous publications these effects were mentioned, but no quantified. The authors careful testing of multiple substrates (particle sizes) allowed selection of conditions where the enhancement was greatest.

The substrate design is excellent. The use of reverse micelles, block polymers and subsequent growth of the gold nanoparticles can be performed over large areas at modest cost. The use of gold, rather than more commonly used silver, will allow the nanoparticles substrates to be stable for long periods of time.

Reply. We are very grateful to the reviewer for her/his enthusiastic comment on our study. We appreciate such a motivated encouragement to realize an actual device for diagnosing malaria as well as other diseases.

There is one point which the authors can make move clear. What is the ???? between the 7×10^4 enhancement on page 10 and eq (1) on pH. The selection of $F_2 = 190$ is not clear.

Reply. We assume the Reviewer is asking more details about the relation between the enhancement factor G and the parameter F_2 coming out from the best fit of the experimental data with equation (1). In this regard, we point out that the fluorescence intensity shown in figure 4(c) is proportional to G , and, hence, the fluorescence enhancement plays a key role in making the signal detectable, the latter being provided by a sCMOS camera in arbitrary units. In order to ease the readability of the plots, the original values of the fluorescence intensity were scaled down by an arbitrary factor so to have the minimum intensities in the first two decades (0.1-10). Thus, while F_2 value is somehow proportional to G , we dealt with its scaled value, which was returned by the best fit procedure as 190 ± 10 arb. units.

This is now clarified in the manuscript by adding the following sentence to the paragraph **Processing and analysis of the fluorescence images.**

“To ease the readability of the calibration curve (Figure 4c), the intensities were scaled down by an arbitrary factor.”

Response to Reviewer #2 (in *red italics the reviewer's comment*).

Reviewer #2: In this manuscript the authors report on a novel and very sensitive technique to measure malaria biomarkers. They make use of a plasmonic enhancement technique in order to significantly increase the fluorescence intensity, together with an aptamer which increases specificity to the target biomarker. The approach which they use is applicable to a wide range of different targets and so I believe that this paper would be of interest to a general audience in the field of biosensing. The authors describe the experimental methods in sufficient detail to permit reproducibility and make reference to appropriate previous work. They also provide extensive supplemental studies which support their main conclusions and which provide more details that will be of interest to readers. The level of sensitivity and specificity which is achieved is very impressive and is superior to currently used methods. Therefore, I believe that this manuscript should be published in Nature Communications. I do however have a number of comments and questions that should be addressed by the authors.

Reply. We are very grateful to the reviewer for his very positive opinion about our manuscript and appreciate very much the encouragement to extend the application of our method.

1. Based on the electromagnetic analysis presented in figure 2(d) the authors choose a particle diameter of 50 nm and interparticle distance of 20 nm, claiming that will give the greatest fluorescence enhancement. However, in figure 3 (f) they show that the extinction spectrum of a substrate fabricated at those dimensions would lie outside the absorption and emission peak of the fluorescent dye. It is only due to fabrication imperfections that the extinction peak maximum is coincident with the absorption and emission peaks. Clearly, as demonstrated by their experimental work there is a fluorescent enhancement. However, have they should provide better explanation as to whether the 50-20 nm spacing is really optimal from a theoretical point of view, and whether the enhancement would still be observed if fabrication had been perfect. This would be important for other researchers seeking to follow up on this work.

Reply. As the Reviewer correctly points out, the plasmon resonance of an ideal lattice may differ from the fabricated one to some extent. Thus, the choice of the fluorophore should be done after the fabrication so to effectively match the fluorophore excitation and emission peaks with the actual plasmon resonance. Such an overlap warrants the occurrence of the fluorescence enhancement since it is the essential condition for the dual-mechanism enhancement. On the other hand, the wide variety of fluorescent dyes apt to this aim allows one to easily fulfil such a requirement. We modified the text that reads as follows (in italics the new sentence).

“The superposition of the plasmonic resonance to both excitation and emission peaks of the fluorophore provides an ideal condition for achieving a PEF amplification since both FRET and Purcell effect can take place^{13,16}. *Such a superposition can be easily realized at essentially any substrate plasmon resonance given the wide variety of fluorescent dyes apt to be used as aptamer label*”.

As it concerns the choice 50-20 nm, it arose from the simulations reported in Figure 2d that suggested the occurrence of higher fields, and hence higher enhancements, when the nanoparticle diameter D is kept small (the ratio R between the diameter and the interparticle distance is fixed at $R=2.5$). This conclusion is still true in the revised manuscript, although Figure 2d required a change based on the Reviewer's comment #2 (see next point). For the sake of clarity, we revised the paragraph in which such a discussion is reported in order to highlight that the 50-20 nm appears to be the optimal choice when $R=2.5$ is considered. Here below the new paragraph.

“A schematic representation of the simulation workspace and unit cell are shown in Figures 2a and 2b, respectively. The E-field intensity exhibits a maximum in the $x - y$ plane along the polarization direction, while it shows a minimum in the transverse direction (see Supplementary Figure S4). By rotating the direction of polarization around z , we observed an azimuthal E-field intensity modulation with an expected period of 60° . Since all the polarization directions have to be taken into account when non-polarized light is used (e.g. light emitted by a fluorophore), the actual local E-field intensity is the result of the average of all such contributions (see Supplementary Figure S5a). It turns out that the gain factor $|E/E_0|^2$ slightly modulates around a value of 4 when φ is changed from 0° - 360° and the distance is 10 nm from the nanoparticle surface (see Supplementary Figure S5b), thereby suggesting that the E-field intensity experienced by the fluorophore has a relatively weak azimuthal dependence.

The analysis as a function of the polar angle θ is reported in Figure 2c that shows the distribution of the E-field intensity in the $y = 0$ plane, resulting from the interaction of an x -polarized plane wave with a 2D honeycomb lattice (Figure 2b) made of 50 nm AuNPs distanced 20 nm one from the other. Due to the plasmon coupling, the largest values of the field intensity are recorded along the direction of polarization (x) while the smallest values are measured in the transverse direction to the plane $z = 0$. In particular, the gain is more effective in the interval $45^\circ \leq \theta \leq 135^\circ$ that contains approximately 70% of the available surface.

Aiming at optimizing the performance of the substrate, we conducted a test-campaign where the response of the system was analysed as a function of the characteristics of the 2D lattice. In the simulations, we varied the particle size and interparticle distance over large intervals ($25\text{ nm}/10\text{ nm}$, $50\text{ nm}/20\text{ nm}$, $75\text{ nm}/30\text{ nm}$, $100\text{ nm}/40\text{ nm}$) keeping constant their ratio $R = 2.5$, a value that is large enough to warrant a high enhancement factor. Figure 2d shows the enhancement of the E-field intensity for several polar angles ($\theta = 0^\circ, 30^\circ, 60^\circ, 90^\circ$) at a distance of 10 nm from the AuNP surface (points P₁-P₄ in Figure 2c). Despite the strongest gain was achieved for the smallest nanoparticles, the need to allow the formation of the Ab-PfLDH-Apt* sandwiches in the interparticle gaps suggested us to consider $50\text{ nm}/20\text{ nm}$ as the optimal choice to guarantee good enhancement of the E-field intensity while assuring that the process of functionalization and detection occurs correctly.”

2. It is assumed throughout (and stated on page 4) that the nanoparticles are hemispherical in shape. However, the highest magnification image (Fig 3(b)) does not provide enough detail to allow the reader to see whether that assumption is justified. The authors should provide a better analysis of the shape of the nanoparticles.

Reply. We thank the Reviewer for raising such an issue that allowed us to realize that the actual nanoparticle shape is spherical rather than hemispherical. Actually, the fabrication technique employed in this study, in which gold nanoparticles are obtained from a seed initially deposited on a substrate, leads to particles with a *spherical* shape, as evidenced by several publications in the field¹⁻⁵, the first of which is cited in the manuscript. We revised the simulations of an ideal lattice to keep into account the correct shape of the nanoparticles finding that new Figure 2d still allows us to asses that the best trade-off takes place with 50-20 nm geometry.

Additionally, we took this opportunity to significantly improve the analysis of the optical response of the 2D nanoparticle array by considering the *actual* morphology resulting from SEM rather than an ideal lattice. Quite remarkably, the simulated extinction spectrum and the experimental one are in very good agreement. We highlighted this in the revised manuscript as reported below.

“The experimental extinction spectrum is well reproduced by that one worked out by considering the real morphology of the substrate as provided by SEM (gold continuous line), the latter being obtained by

averaging the extinction spectra resulting from 15 regions of interest sampled from the SEM image shown in Supplementary Figure S7a (an example of the rendering provided by Lumerical is shown in Supplementary Figure S7b, whereas some of the simulated extinction spectra are reported in Supplementary Figure S7c).”

The information about the importing of the SEM image into Lumerical software were included in the section Methods.

1. Lee, W., Lee, S. Y., Briber, R. M. & Rabin, O. Self-Assembled SERS Substrates with Tunable Surface Plasmon Resonances. *Adv. Funct. Mater.* **21**, 3424–3429 (2011).
2. Lemineur, J.-F. & Ritcey, A. M. Controlled Growth of Gold Nanoparticles Preorganized in Langmuir–Blodgett Monolayers. *Langmuir* **32**, 12056–12066 (2016).
3. Chen, J. *et al.* Scalable Fabrication of Multiplexed Plasmonic Nanoparticle Structures Based on AFM Lithography. *Small* **12**, 5818–5825 (2016).
4. Zhang, H. *et al.* Arbitrary Gold Nanoparticle Arrays Fabricated through AFM Nanoxerography and Interfacial Seeded Growth. *ACS Appl. Mater. Interfaces* **11**, 38347–38352 (2019).
5. Lu, H. *et al.* Ag nano-assemblies on Si surface via CTAB-assisted galvanic reaction for sensitive and reliable surface-enhanced Raman scattering detection. *Sensors Actuators B Chem.* **304**, 127224 (2020).

3. On Page 3, the authors point out that pre-functionalized biosensors for malaria often have transportation and storage difficulties. In the discussion section they should justify whether their approach reduces these difficulties.

Reply. We thank the reviewer for this observation that allowed us to stress the role played by our functionalization technique. Actually, although simple in the usage, the conventional rapid test cannot be functionalized on-site since they are designed to be pre-packaged and no part of the fabrication can be carried out outside the production site. On the contrary, due to its simplicity and rapidity, our functionalization procedure can be realized at a Point-Of-Care site by the personnel available there. Thus, the plasmonic substrate can be delivered separately from antibodies and aptamers for which standard delivering and storage procedures are available. The possibility to functionalize the substrate on-site without any expensive instrumentation and laborious procedure (PIT can be carried out by a simple UV lamp in a few minutes) reduces the need for storing at low temperature to the small volumes required by the vials containing the biomolecules.

To stress the point raised by the Reviewer, we have added two sentences to the first paragraph of the Discussion to clarify that our approach may be able to reduce the cold transport challenges for rapid diagnostic tests for malaria. We further expand in the new second paragraph of discussion how our approach may allow saliva based non-invasive diagnosis of malaria.

4. On page 6 the authors appear to find it remarkable that their periodic structure would have an azimuthal E-field modulation with a period of 60 degrees. However, given the hexagonal symmetry of the nanostructure that they simulate, this does not seem particularly remarkable to me.

Reply. We fully agree with the Reviewer and this sentence was revised (see the answer to comment #1).

5. On page 7, the authors state ‘At all considered points, the enhancement decreases with the diameter of the AuNP, with the exception of P1 with an increment for $D=50$ nm slightly larger than that for $D=25$ nm’. What does that mean?

Reply. This sentence was fully revised (see the answer to comment #1).

6. Figure 3 (d) and (e) shows the lattice spacing measured from the micrographs. The authors should provide more details as to how these spacings were calculated from the images.

Reply. The substrate was observed by Zeiss LEO 1550VP field emission scanning electron microscope (FESEM) with a nominal resolution of 1 nm at 20 kV acceleration voltage. The recorded SEM images (Supplementary Figure S12a shows an example of raw image at high magnification) were processed by ImageJ software to retrieve information about the substrate morphology. To this aim, each image was binarized to isolate the objects from the background (Supplementary Figure S12b) and segmented by “Watershed” tool implemented in ImageJ to separate adjacent nanoparticles (Supplementary Figures S12c). Then, object area S , perimeter p , shape descriptors (aspect ratio AR and circularity $4\pi S/p^2$) and centroid coordinates were measured by “Analyze Particles” tool implemented in ImageJ. Supplementary Figure S12d shows an example of processed SEM image, in which the objects are decomposed in outlines (black line) and inner area (orange filling). Given the round shape of the objects (Supplementary Figure S13), nanoparticle diameter was estimated as $D = 2\sqrt{S/\pi}$, whereas the centre-to-centre distance distribution was carried out by calculating the distance of each centroid from its nearest neighbours.

The above paragraph is now a new subsection in **Method**.

7. Figure 4(b) captions refers to the sample as being ‘contaminated’ human blood. ‘Spiked’ is probably a more suitable word.

Reply. We made the suggested changes.

We wish to thank again the reviewer for his constructive suggestions that allowed us to significantly improve the presentation of our results.

Response to Reviewer #3 (in *red italics* the reviewer's comment).

Reviewer #3: The MS by Minopoli et al entitled "Ultrasensitive antibody-aptamer plasmonic biosensor for malaria biomarker detection in whole blood" describes the development of a plasmon-enhanced fluorescence immunosensor for the specific and ultrasensitive detection (femtomolar level) of Plasmodium falciparum lactate dehydrogenase (PfLDH) in whole blood.

The results are novel (a new immunosensor is described) and of interest to the malaria community but also to other fields working on the development of highly sensitive devices to detect microbial antigens. Given the nature of the work presented, no statistical analysis are included in the paper. I have no comments with regards to the development of the assay, as the procedures are out of my expertise. However, I have two main comments with regard to the validation and use cases of the assay developed.

Reply. We thank the reviewer for his positive comments about our work. We amended the manuscript and included some statements to provide more information about the questions he raised.

1. Which is the expected use of an assay that allows detecting such low amounts of a Pf antigen? Are the authors considering its use for the validation of malaria rapid diagnostic tests (RDT)? For the use in the field? In the first sentence of the last paragraph in the discussion, authors seem to suggest its clinical potential. In which situations such a highly-sensitive test would be used? For what purpose? There have been several discussions in the malaria community about the use of a recently developed RDT which has higher sensitivity than standard RDTs, without achieving a consensus of the rationale for detecting extremely low parasite densities. Costs should be considered in this analysis.

Reply. The approach we describe in this manuscript is over 1000 times as sensitive as the immunochromatographic methods used presently clinically for rapid diagnosis of malaria. An ultrasensitive assay in particular may allow the switch from serum to saliva for malaria diagnosis. We expand on this point in a new paragraph 2 of the discussion.

2. Authors have used recombinant PvLDH spiked in blood to assess the performance of the assay. How was PvLDH produced? Also obtained from bacterial expression? How well did these proteins reflect the native conformation of the antigens? Which is the LOD and the general performance of the assay when tested against PFLDH and PvLDH in clinical samples?.

Reply. PvLDH was also produced by bacterial expression and we have added the relevant citation to Methods section page 14. One of the authors has solved crystal structures of both PFLDH and PvLDH thus we have data to support that these reflect the native conformation of the antigens. A full clinical study is beyond the scope of the present manuscript, but will be a followup. We expand on this in the second paragraph of the discussion.

We wish to thank again the reviewer for his constructive comments that allowed us to highlight the applications of our biosensor.

Reviewer #2 (Remarks to the Author):

The authors have addressed all of my previous concerns and I have no further suggestions to make. This paper is very clearly written and will be of great interest to a wide range of researchers in the field of optical sensing. The experimental results are fully supported by the extensive theoretical and numerical modeling. An outstanding level of sensitivity is obtained. I fully support the publication of the article in Nature Communications.

Reviewer #3 (Remarks to the Author):

I agree with the answers given to my comments. I do not have any further question.

Response to Reviewer #2 (in *red italics* the reviewer's comment).

Reviewer #2: The authors have addressed all of my previous concerns and I have no further suggestions to make. This paper is very clearly written and will be of great interest to a wide range of researchers in the field of optical sensing. The experimental results are fully supported by the extensive theoretical and numerical modeling. An outstanding level of sensitivity is obtained. I fully support the publication of the article in Nature Communications.

Reply. We are very grateful to the reviewer for his very positive opinion about our paper and sincerely thank him for his work: The quality of the presentation significantly improved thanks to his clever and constructive comments.

Response to Reviewer #3 (in *red italics* the reviewer's comment).

Reviewer #3: I agree with the answers given to my comments. I do not have any further question.

Reply. We wish to thank the reviewer once more for his work that allowed us to improve the quality of the paper.